# Numerical Analysis of Ground Surcharge Effects on Deformation Characteristics in Shield Tunnel Linings

Lixin Wei [1], Chunshan Yang [1], Weijie Chen [2,*], Liying Liu [1] and Dong Su [2,3,4,*]

[1] Guangzhou Municipal Engineering Design & Research Institute Co., Ltd., Guangzhou 510030, China; wlxgxm@163.com (L.W.); soildoctor@163.com (C.Y.); jaysnowing@163.com (L.L.)
[2] College of Civil and Transportation Engineering, Shenzhen University, Shenzhen 518060, China
[3] Key Laboratory for Resilient Infrastructures of Coastal Cities (MOE), Shenzhen University, Shenzhen 518060, China
[4] Shenzhen Key Laboratory of Green, Efficient and Intelligent Construction of Underground Metro Station, Shenzhen 518060, China
* Correspondence: 2250471016@email.szu.edu.cn (W.C.); sudong@szu.edu.cn (D.S.)

**Abstract:** To investigate the deformation characteristics of shield tunnel linings under ground surcharge, finite element software was employed to create a detailed three-dimensional model of the staggered assembly of the shield tunnel lining. This model includes components such as precast concrete segments, reinforcements, and joints (comprising bent bolts, washers, and bolt sleeves). Additionally, the model accounts for interface frictions between segments and the interactions between different rings. The reliability of the numerical model was verified based on the results of a full-scale model test. Additionally, the model accounts for interface frictions between segments and the interactions between different rings. Changes in tunnel convergence, joint tensioning, bolt stresses, reinforcement stresses, and concrete crack development were systematically analyzed. The results indicate the following: (1) the deformation mode of the lining structure under ground surcharge resembles a "transverse ellipse". Joints located near the haunch opened along the outer arc, while those near the vault and bottom opened along the inner arc. The restraining effect of the bolts on joints opening in the inner arc was greater than that on the outer arc. Notably, when the opening of the inner arc reached 4.9 mm, the bolt stress escalated to the yield strength of 640 Mpa. (2) Under larger loads, the lining structure's joints are susceptible to greater deformation, resulting in the tensile yielding of local reinforcement within these joints. (3) Cracks predominantly occur near the haunch, vault, and bottom of the lining structure, with the central angle of crack distribution ranging between 70° and 85°.

**Keywords:** ground surcharge; shield tunnel; lining structure; deformation characteristics; numerical analysis

## 1. Introduction

With the burgeoning development of urban areas and the continuous increase in population density, the daily commuting volume is gradually rising, leading to increasingly strained surface transportation. To alleviate surface traffic congestion, the expansion of underground transportation networks has become essential. As vital components of metropolitan infrastructures, subways and road tunnels play a crucial role in urban mobility [1]. The shield method, notable for its efficiency, cost-effectiveness, and minimal environmental impact, has emerged as the predominant construction technique for these structures [2]. However, the integration of new construction projects poses significant challenges to existing tunnels, such as the impact of ground surcharge and the unloading of excavation pits adjacent to construction loads. Under the influence of these new construction loads, the structural force balance of tunnels can be disrupted, leading to excessive convergence, joint opening, and cracking in lining structures [3–6]. In major

cities like Shanghai, ground surcharge impacts, primarily from new constructions or soil accumulation in foundation pits, have been identified as primary causes of operational disruptions. And these impacts often result in substantial deformation and joint openings in tunnels, threatening their operational viability [3,7]. This study aims to address these challenges by investigating the effects of ground surcharge on tunnel structures, with a focus on deformation characteristics.

The impact of ground surcharge on tunnel structures has been extensively explored in past research, with the use of diverse methods, such as theoretical calculations [3,8–11], full-scale [12–14] and scaled model tests [15–17], field measurements [18,19], and, predominantly, numerical simulations. The preference for numerical simulations stems from their cost-effectiveness and capability to yield comprehensive data. This study seeks to advance this field by addressing the limitations of the existing approaches and enhancing the precision and applicability of simulation methodologies, particularly in the context of complex urban tunnel systems. This study establishes a three-dimensional refined analysis model of a shield tunnel lining structure with staggered joint assembly using finite simulation calculations, including precast concrete segments, rebar, and joint bolts. It also takes into account the interfacial friction between segments and the interaction between rings, and verifies the reliability of the numerical model based on the results of full-scale model tests.

In recent studies, the stability and deformation of various tunnel types under ground surcharge have been extensively explored. Wang et al. [20] used numerical simulation to study the evolution of the transverse deformation of shield tunnels under ground surcharge, established the relationship between the convergence of the tunnel and the concrete force, bolt force, and joint opening, and the proposed convergence of the tunnel as a judgement indicator for the development of the lateral structural state of tunnels. Huang et al. [21] developed a 3D numerical model of a shallow shield tunnel in soft soil, assessing ground and tunnel deformations under various surcharge scenarios. The numerical simulations show that the surface and tunnel deformation responses vary when induced by different loading modes above shallow shield tunnels in soft soil. Prassetyo et al. [22] created two tunnel models in different permeability soils using Fast Lagrangian Analysis of Continua, focusing on surcharge effects in saturated foundations, and noted that the stability of tunnels in saturated foundations is mainly affected by the permeability of the lining and the long-term hydraulic response of the foundations. Zhang et al. [23] investigated the coupled effect of the soil spatial variations and the disturbance of ground surface on the existing tunnel using the random finite difference method. It demonstrated that neglecting spatial variability will cause an underestimation of the tunnel convergence, especially when it is severely disturbed by nearby environments, such as the large surface surcharge. Wei et al. [24] established a 3D analytical model of a shield tunnel to examine longitudinal deformation under surcharge, comparing these findings with experimental results. The results showed that the greater the ground surcharge, the greater the deformation of the tunnel. The further away from the ground surcharge, the smaller the deformation of the tunnel. Du et al. [25] combined hyperstatic reaction methods and numerical simulations to study horseshoe-shaped tunnels in saturated soft rocks, finding consistency between the two approaches. The results showed that ground surcharge has a significant impact on the lining normal forces when the tunnel buried depth is approximately four times its height. Xiao et al. [26] adopted adaptive finite element limit analysis to investigate single and double circular tunnels at varying depths within cohesive–frictional soils or rock masses. They developed a novel approach that utilizes tunnel stability numbers to assess the stability of these tunnels, providing both upper and lower bounds for these critical values.

Previous research has primarily utilized the stratum structure method to analyze the deformation of shield tunnel linings under ground surcharge, effectively simulating soil-lining interactions. However, lining structures are often simplified by using stiffness reduction methods to represent the lining structure as a homogeneous annulus. Alternatively, linear springs or joint units are employed to simulate the behavior of segment joints,

or the interaction between bolts and segments is simplified by embedding bolts within the concrete of the segments. These approaches fail to accurately reflect the true mechanical behavior of joints, and most numerical simulations overlook the role of reinforcement. Moreover, most research does not take into account the interaction between lining rings, which makes it challenging to accurately represent the mechanical behavior of shield tunnel linings assembled with staggered joints. Alternatively, the loading structure method offers a more streamlined approach by representing soil complexities with equivalent loading, enhancing computational efficiency. This study proposes leveraging this method to develop a sophisticated finite element model of shield tunnel linings. Our aim is to investigate in greater depth the deformation characteristics under ground surcharge, addressing the limitations of current methodologies and providing a more comprehensive understanding of structural responses in these critical urban infrastructures.

## 2. Three-Dimensional Finite Element Modelling and Validation

### 2.1. Shield Tunnel Linings

This study focuses on the assembly of shield tunnel lining rings, comprising six precast concrete segments with staggered joints for enhanced structural integrity. Specifically, the assembly includes diverse segments—key (F), adjacent (L1, L2), and standard (B1, B2, B3)—with distinct central angles (15°, 64.5°, and 72°) forming a ring of 6000 mm diameter, 300 mm thickness, and 1500 mm width. The dimensions of the lining structure, the number of lining rings, and the assembly were selected based on full-scale model tests in this study. Detailed segment configurations are illustrated in Figure 1a–c. The connections between segments are secured with M24 high-strength bolts, rated at grade 8.8, arranged in pairs along the circumferential joints, resulting in twelve bolts per ring. Longitudinally, bolts are spaced every 36°, totaling ten per ring. The interfaces between segments, both circumferentially and longitudinally, are flat. Bolts and nuts measure 24 mm and 36 mm in diameter, respectively, with washers at 64 mm in diameter, as illustrated in Figure 1d.

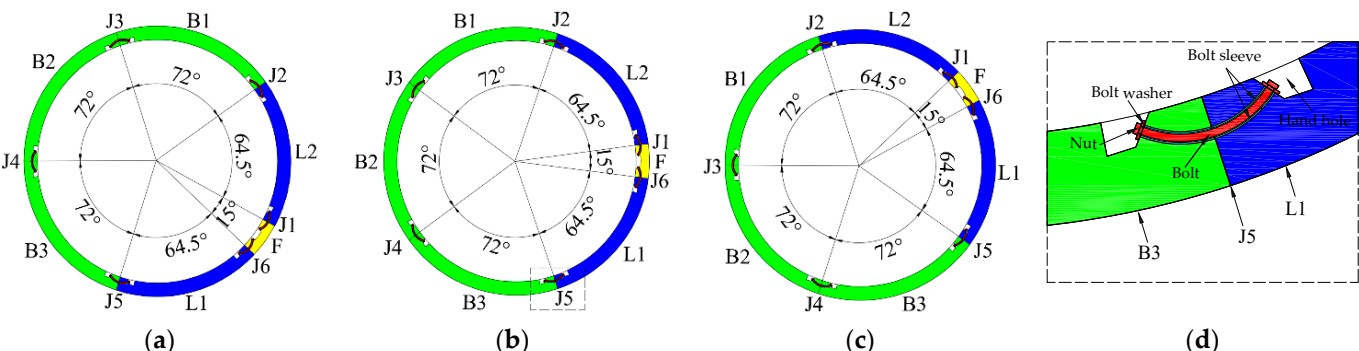

**Figure 1.** Schematic diagram of lining blocking and joint construction. (**a**) Ring 1; (**b**) Ring 2; (**c**) Ring 3; (**d**) joint construction.

### 2.2. Finite Element Model

This analysis presents a finite element model of staggered shield tunnel linings, incorporating elements such as precast concrete segments, bent bolts, bolt washers, bolt sleeves, and reinforcement. The reinforcement's quantity and distribution mirror the actual setup of the precast concrete segments. The finite element model, showcasing the staggered arrangement in the shield tunnel lining, is illustrated in Figure 2. Figure 2a presents a schematic of the finite element model for the full-ring structure. The numerical model in this study was established based on a selection derived from full-scale model tests, consisting of two semi-cut lining rings (each with a width of 750 mm) and one complete lining ring (with a width of 1500 mm), with the central ring being the complete one. The primary focus of the full-scale tests is on the transverse deformation characteristics of the lining, with a special emphasis on the deformation properties of the full-ring lining structure [12]. The inclusion

of two semi-cut lining rings is designed to consider the boundary conditions of misaligned assembly in the complete lining ring. Since this study primarily investigates the impact of ground surcharge on the transverse deformation of the lining ring and also takes into account the complexity of a refined finite element model of the lining structure, the effects of using a complete lining ring in place of two semi-cut lining rings for the misaligned assembly boundary conditions are fundamentally consistent. To avoid the need for remodeling semi-cut lining rings, the finite element model also utilizes complete lining rings for the semi-cut lining rings on both sides. Figure 2b illustrates the finite element model of the segmental structure, which includes circumferential and longitudinal bolt access holes for the positioning of circumferential and longitudinal bolts, respectively. To achieve precise deformation simulations, our model employs reduced integration eight-node linear elements (C3D8R), optimizing the accuracy and computational efficiency. These elements are crucial for representing concrete segments, bolts, washers, and sleeves, as they allow for the detailed analysis of stress distribution and deformation under load. Similarly, the use of three-node truss elements (T3D3) for simulating reinforcement is vital for accurately capturing the behavior of structural steel within concrete, reflecting its contribution to the overall structural integrity. For instance, in a simulated load test, C3D8R elements enabled us to observe the segments, while T3D3 elements illustrated how reinforcement bars absorbed tension forces.

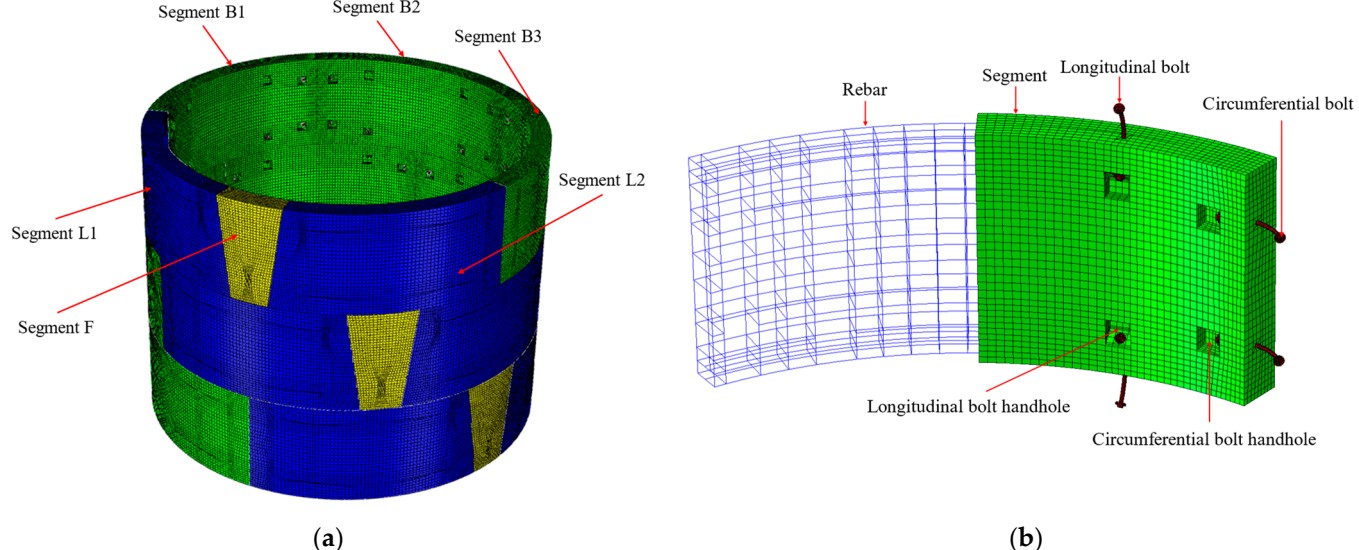

**Figure 2.** Finite element model. (**a**) Full ring; (**b**) segment.

The structural and finite element models developed in this study are based on a three-ring staggered assembly of tunnel segments. This design aims to mimic the boundary constraints of staggered assemblies, although some differences from actual conditions may persist. In practical engineering, shield tunnels consist of multiple rings of segments that are more adept at managing deformation than models with fewer rings, potentially resulting in a symmetrical deformation pattern. Despite this, models with a single ring or three rings serve as vital representations of localized sections within shield tunnels, capturing essential aspects of their behavior. These models are particularly valuable for understanding the deformation and stress responses of segment structures, offering high reliability in these aspects.

### 2.3. Constitutive Model and Parameters

This study employs the Concrete Damage Plasticity (CDP) constitutive model, initially proposed by Lubliner [27], to simulate the mechanical behavior of concrete. This model simulates the formation and propagation of microcracks in concrete under load by introducing

a damage variable, thus enabling a more accurate prediction of the mechanical properties and failure process of concrete. Specifically, the Concrete Damage Plasticity (CDP) model divides the stress–strain relationship of concrete into elastic, plastic, and damage phases. In the elastic phase, concrete exhibits linear elastic behavior. As stress increases, concrete enters the plastic phase, beginning to undergo irreversible deformations. Upon reaching a certain stress level, concrete moves into the damage phase, where microcracks start to form and propagate, leading to a gradual reduction in the concrete's strength and stiffness. The choice of the CDP model to simulate the mechanical behavior of concrete was primarily due to its ability to accurately capture the nonlinear behavior and damage evolution of concrete under complex stress states. This is of significant importance for our study of the failure mechanisms of concrete structures, optimization of design, and enhancement of structural safety. Widely used in reinforced concrete structure research, the CDP model comprehensively covers both the tension and compression behavior of concrete, including its damage factors. Figure 3 shows the compressive damage and tensile damage behaviorof the concrete C50. The concrete's stress–strain relationship is determined in line with the Code for Design of Concrete Structures (GB50010-2010) [28]. Additionally, the concrete damage plasticity model's damage factor is calculated based on Sidoroff's energy equivalence principle [29]. During the full-scale model experiments, Wang and Lu [12,13] tested the compressive strength of concrete, determining it to be 50 MPa. However, the tensile strength of concrete was not assessed. Thus, this study utilized the formula provided by the standards for calculating the tensile strength of concrete from its compressive strength. According to the currently utilized Building Code Requirements for Structural Concrete and Commentary (ACI 318-99) [30] in the United States, the concrete tensile strength ($f_t$) can be calculated from the compressive strength ($f_c$) using the formula ($f_t = 0.56 f_c^{0.5}$), resulting in $f_t$ = 3.96 MPa. Table 1 lists the parameters of CDP model.

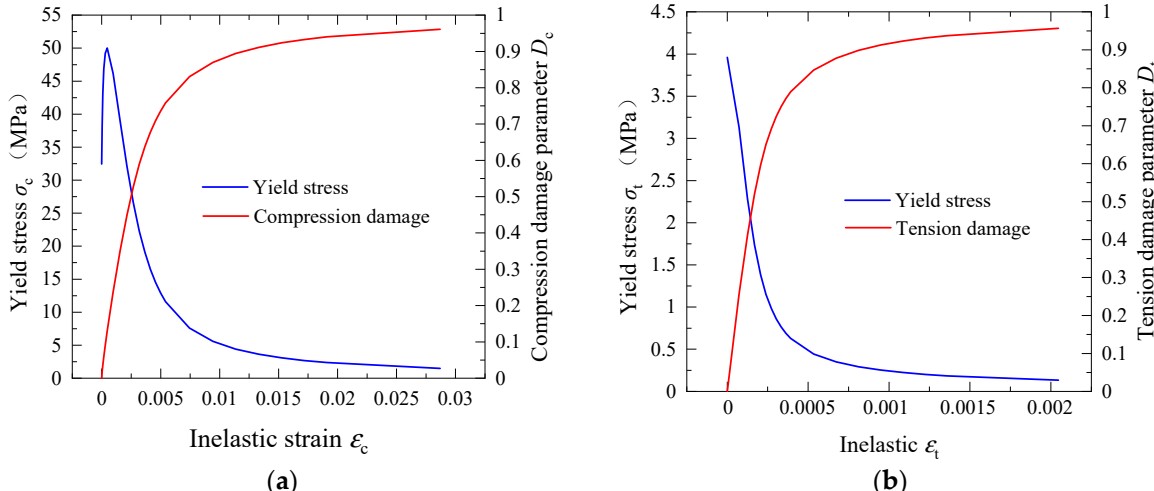

**Figure 3.** Damage behavior of the concrete C50. (**a**) Compression damage; (**b**) tension damage.

**Table 1.** Parameters of the Concrete Damaged Plasticity model for the concrete.

| Parameters | Values | Parameters | Values |
|---|---|---|---|
| Mass density $\rho$ (kg/m$^3$) | 2500 | $f_{b0}/f_{c0}$ | 1.16 |
| Young's modulus $E_c$ (MPa) | 34,500 | $e$ | 0.1 |
| Poisson's ratio $\nu$ | 0.2 | $\sigma_{cu}$ (MPa) | 50 |
| Dilation angle $\psi$ (°) | 30 | $\varepsilon_{cu}$ | 0.00192 |
| Invariant stress ratio $K_c$ | 0.6667 | $\sigma_{t0}$ (MPa) | 3.96 |
| Viscosity parameter $\mu$ | 0.0005 | $\varepsilon_{t0}$ | 0.000137 |

Note: $f_{b0}/f_{c0}$ is the biaxial/uniaxial compression plastic strain ratio; $e$ is the flow potential eccentricity; $\sigma_{cu}$ and $\varepsilon_{cu}$ are the axial compressive strength and the corresponding strain (the peak compressive strain); $\sigma_{t0}$ and $\varepsilon_{t0}$ are the axial tensile strength and the corresponding strain (the peak tensile strain), respectively.

In this model, both the reinforcement and bolts exhibit elastic–plastic bi-polyline behavior, whereas the bolt washers and sleeves are modeled with linear elasticity. The elastic modulus and Poisson's ratio for the reinforcement, bolts, and washers are uniformly 206,000 MPa and 0.3. In contrast, the bolt sleeves have an elastic modulus of 2000 MPa and a Poisson's ratio of 0.35. According to the Code for the Design of Concrete Structures (GB 50010–2010), the standard value of the yield strength and the ultimate strength of HPB400 reinforcement are 400 MPa and 500 MPa, respectively. The grade of bolts is 8.8, of which the yield strength and ultimate strength are 640 MPa and 800 MPa, respectively.

*2.4. Interactions*

This study's finite element model effectively captures the friction at the interface between segments. It models the interactions between adjacent segments and the connections between bolts and bolt sleeves through surface-to-surface contact. In the finite element software, this contact is represented either as interactions between two deformable surfaces or between a deformable and a rigid surface, as per reference [31]. The finite slip tracking method and surface-to-surface discretization are employed in this study. This method can effectively simulate the sliding friction between two surfaces. The friction coefficients are set at 0.5 for adjacent segments and 0.3 for bolts and bolt sleeves. Interactions between rebars and concrete segments are modeled using embedded constraints. Tie constraints are utilized for connections between the nut and bolt washer, the bolt washer and concrete, and the bolt sleeve and concrete [32]. Figure 4 presents a schematic illustration of the interactions among the various components. During the process of shield tunnel forces, the interactions include both the circumferential and longitudinal segments. The interactions between longitudinal segments and between circumferential segments are identical. These interactions encompass contact between segments, between washers and segments, between nuts and washers, and between bolts and bolt sleeves, as well as between bolt sleeves and segments.

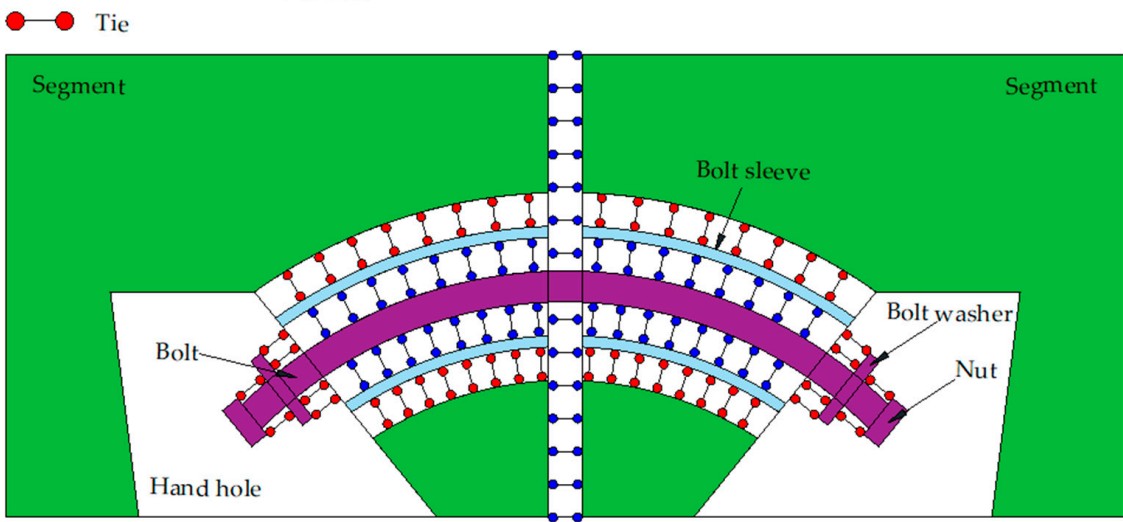

**Figure 4.** Schematic diagram of interactions.

*2.5. Validation*

Wang et al. [12] conducted full-scale model tests on shield tunnel lining structures, as illustrated in Figure 5a, investigating the deformation characteristics and failure mechanisms of these structures under various load conditions. The dimensions of the lining structure, the number of lining rings, and the assembly were selected based on full-scale model tests in this study. This study aimed to enhance the verification of the numerical model's accuracy; thus, the distribution and magnitude of loads in the numerical sim-

ulation were directly based on the experimental setup, as illustrated in Figure 5b. The loading method of the experimental loading device is a concentrated load application, utilizing 24 concentrated loads to simulate the earth pressure on the tunnel, as illustrated in Figure 5c. $P_1$ is primarily employed to replicate the earth pressure at the top of the tunnel, $P_2$ simulates the lateral earth pressure on the tunnel, and $P_3$ represents the earth pressure on the tunnel's shoulder transition section. Table 2 lists the load magnitudes in the segment full-scale model test. Two sets of jacks and corresponding loading beams were used to apply each load on the segments, while half-ring segments were loaded with a single set of jacks and a loading beam. The segments experienced a longitudinal load of 1500 kN, distributed evenly using 16 jacks. The numerical model replicates a three-ring staggered shield tunnel segment, similar to the full-scale test in terms of the circumferential load application. It also simulates an equivalent longitudinal load distribution of 280 kPa acting on the segment's longitudinal surface.

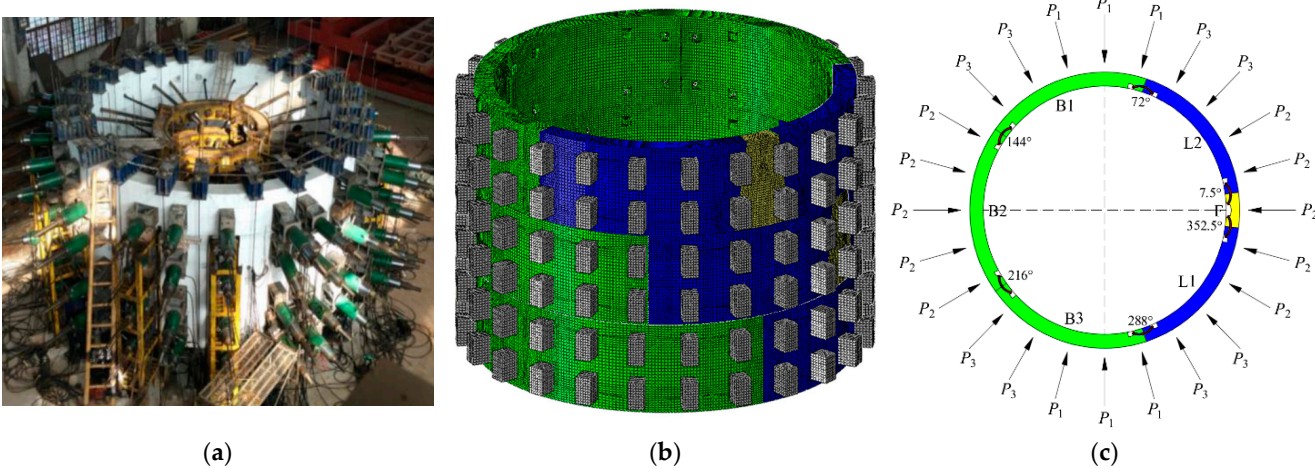

| (a) | (b) | (c) |

**Figure 5.** Schematic diagram of segment full-scale model's test loading. (**a**) Segment full-scale model test; (**b**) distribution of loading beam; (**c**) distribution of test loading.

**Table 2.** The loading procedure in the segment full-scale model test.

| Conditions | $P_1$ (kN) | $P_2$ (kN) | $P_3$ (kN) |
| :---: | :---: | :---: | :---: |
| L0 | 0 | 0 | 0 |
| L1 | 356 | 255 | 270 |
| L2 | 610 | 427 | 445 |
| L3 | 750 | 530 | 558 |
| L4 | 746 | 414 | 436 |
| L5 | 450 | 310 | 320 |
| L6 | 434 | 450 | 436 |
| L7 | 293 | 130 | 167 |
| L8 | 413 | 278 | 283 |
| L9 | 413 | 148 | 220 |

Figure 6 shows the comparison of tunnel convergence between the numerical simulation and full-scale tests. The comparison reveals that the simulation closely aligns with the test results for loading conditions L1 to L3. Discrepancies between the numerical calculations and full-scale experimental results for conditions L3 to L4 are acknowledged. Conditions L1 through L4 represent the loading stages, and, as illustrated in Figure 6, the lining structure's elastic deformation stages from L1 to L3 exhibit a high degree of agreement between the numerical simulations and experiments. This concurrence suggests that the boundary conditions and the elastic mechanical parameters of the materials used in the numerical simulations are consistent with those in the experiments. However, during the transition from L3 to L4, which marks the plastic deformation stage of the lining structure,

the discrepancy between the numerical simulations and experimental results begins to widen. The primary factors in this discrepancy are identified as the plastic mechanical parameters of the concrete and the constitutive model's simulation of concrete's plastic mechanical development. This study employs the Concrete Damaged Plasticity (CDP) model, which, to a certain extent, accurately simulates the plastic characteristics of concrete. Nonetheless, it fails to replicate issues observed in experiments, such as concrete spalling and detachment, which likely contribute to the main source of error. The error is more significant for conditions L4 to L6, with deviations of 26% and 47% for L5 and L6, respectively. The unloading stages from conditions L4 to L6, where the lining structure has already sustained significant damage, result in a more pronounced occurrence of concrete segments detaching during the unloading process. Additionally, the simulation of concrete's plastic development in numerical simulations may exhibit certain discrepancies from full-scale experiments, leading to a greater rebound deformation of the lining ring's waist section in conditions L4 to L6. Conditions L6 to L9 represent the re-loading stages, and, from a trend perspective, there is a relatively high degree of conformity between the numerical simulations and experiments during this phase. The primary cause of discrepancies in this stage is the excessive rebound amount in the lining structure under condition L6. Overall, after the lining structure undergoes plastic deformation, the errors in numerical simulations begin to increase, with the simulation of concrete's plastic development being the primary contributor to these discrepancies. During the elastic loading phase (L1–L3), the numerical simulation showed an approximate error of 5%. In the plastic loading phase (L3–L4), the maximum error reached 26%, while in the plastic unloading phase (L4–L6), it escalated to 47%. In the plastic reloading phase (L6–L9), the error was observed to be 25%.

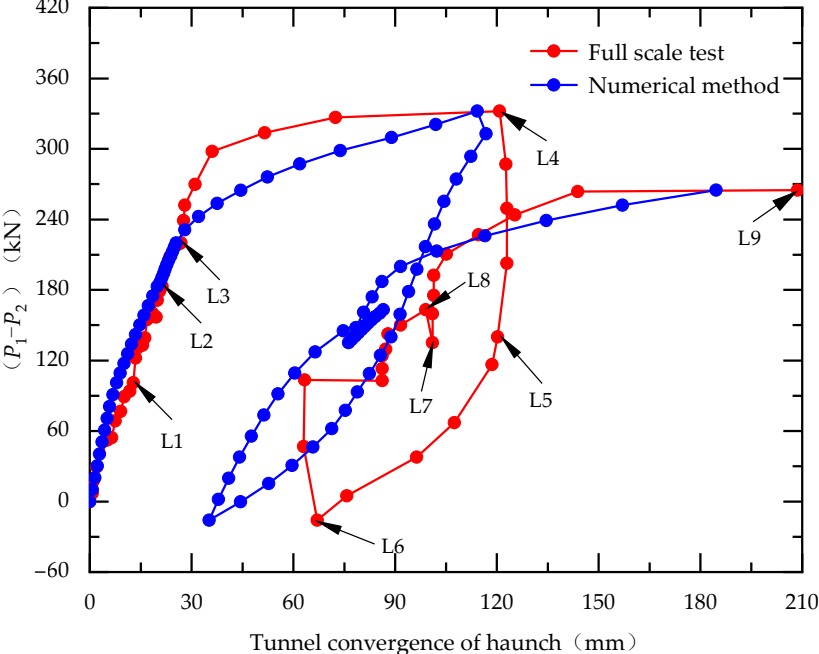

**Figure 6.** Comparison of tunnel convergence from the numerical analyses and the full-scale test.

In this study, a finite element model grounded in a three-ring lining structure is developed. However, there is a gap between this model and the multi-ring lining structures in real engineering, especially concerning boundary conditions. Nevertheless, when juxtaposed with data from comprehensive full-scale tests, this model has proven capable of accurately simulating the interactions between segments. This validation suggests that the model mirrors the mechanical behaviors of single- and three-ring lining structures effectively. Despite some discrepancies, the overall trend in the haunch convergence of the

lining ring is consistent between the simulation and the test, indicating that the simulation reliably represents the deformation and mechanical response of the lining structure.

## 3. Numerical Method and Loading Mode

### 3.1. Loading Mode of Ground Surcharge

During shield tunnel operation, the primary loads include the initial earth pressure from the surrounding ground and an additional load due to ground surcharge. The Boussinesq formula is a mathematical formula applied within the theory of elastic half-space, primarily used for predicting the stress distribution on the ground subjected to point loads. The fundamental assumption of the Boussinesq formula is that the soil is considered an infinite, homogeneous, isotropic elastic half-space. When a point load is applied to the surface of the soil, this formula enables the calculation of the vertical and horizontal stress at any given point. Consequently, this method facilitates the computation of additional loads on tunnels resulting from ground surcharge. As illustrated in Figure 7, the schematic represents a semi-infinite space subjected to a vertical concentrated force (*F*). Upon the application of a vertical concentrated force on the surface of a semi-infinite space, an additional stress is generated at point *M* within the semi-infinite space. The horizontal additional stress, $\sigma_x$, and the vertical additional stress, $\sigma_z$, can be calculated using Equation (1) and Equation (2), respectively.

$$\sigma_x = \frac{3F}{2\pi}\left(\frac{x^2 z}{R^5} + \frac{(1-2\nu)\left[R^2 - z(R+z)\right]}{3R(R+z)} - \frac{(1-2\nu)(x-\varepsilon)^2(2R+z)}{3R^3(R+z)^2}\right) \tag{1}$$

$$\sigma_z = \frac{3F}{2\pi}\frac{z^3}{R^5} \tag{2}$$

$$r = \sqrt{x^2 + y^2 + z^2} \tag{3}$$

where $\nu$ is the Poisson's ratio of the soil and $r$ is the distance from the point of concentration to point *M*.

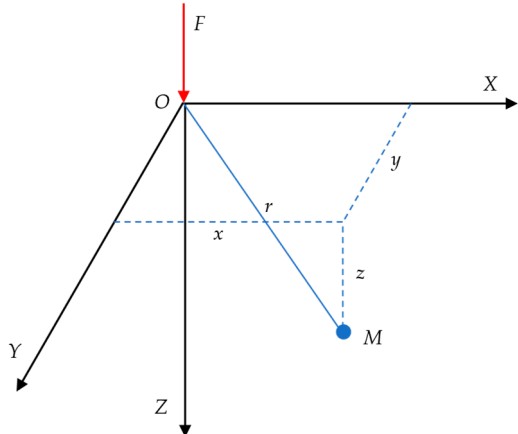

**Figure 7.** The schematic diagram of the semi-infinite space subjected to a vertical concentrated force.

Based on Boussinesq's formula, a calculation model for the additional loads on tunnels due to surface surcharge can be established. Figure 8 presents a scenario where the surface of the operating tunnel is set up with rectangular ground attachments, whose length and width are *L* and *B*, respectively. The horizontal distance from the center of this surcharge to the tunnel's axis is *S*, while the magnitude of the ground surcharge is $P_0$, and the tunnel depth is *H*. Considering the center of the ground surcharge as the origin, the directions parallel and perpendicular to the tunnel are defined as the *y*-axis and *x*-axis, respectively. We assume that a minute element within the range of surface surcharge has a load of

$P_0 d\varepsilon d\eta$, with the center coordinates of the element being $(\varepsilon, \eta, \xi)$, and a certain point on the tunnel surface having the coordinates $(x_0, y_0, z_0)$. According to Boussinesq's formula, the calculation formula for the additional loads on tunnels due to surface surcharge can be established through integration. The horizontal additional stress $Q_x$ and the vertical additional stress $Q_z$ can be calculated, respectively, according to Formulas (4) and (5).

$$Q_x = \frac{3P_0}{2\pi} \iint_\Gamma \left[ \frac{(x_0-\varepsilon)^2 H_0}{R^5} + \frac{(1-2\nu)\left[R^2 - z_0(R+z_0)\right]}{3R^3(R+z_0)} - \frac{(1-2\nu)(x_0-\varepsilon)^2(2R+z_0)}{3R^3(R+z_0)^2} \right] d\varepsilon d\eta \tag{4}$$

$$Q_z = \frac{3P_0}{2\pi} \iint_\Gamma \frac{z_0^3}{R^5} d\varepsilon d\eta \tag{5}$$

$$R = \sqrt{(x_0-\varepsilon)^2 + (y_0-\eta)^2 + (z_0-\xi)^2} \tag{6}$$

where $\Gamma$ ranges from $-B/2 \le \varepsilon \le B/2$, $-L/2 \ \eta \le L/2$, $\nu$ is the Poisson's ratio of the soil, and $R$ is the distance from the minute element to the calculation point.

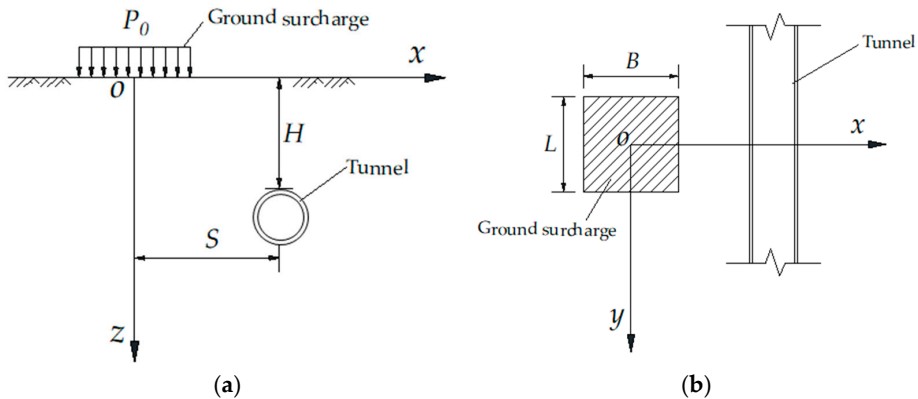

**Figure 8.** Schematic diagram of the ground surcharge calculation model. (**a**) Cross section; (**b**) plan.

Figure 9 presents the variation curves of surcharge loads for a specific case study. In Figure 9a, the distribution of the vertical surcharge load is shown for a burial depth of 15 m ($H$ = 15 m). Conversely, Figure 9b illustrates the distribution of horizontal surcharge pressure located 3 m away from the center of the pile ($S$ = 3 m), over a square area with each side measuring 18 m ($B = L = 18$ m).

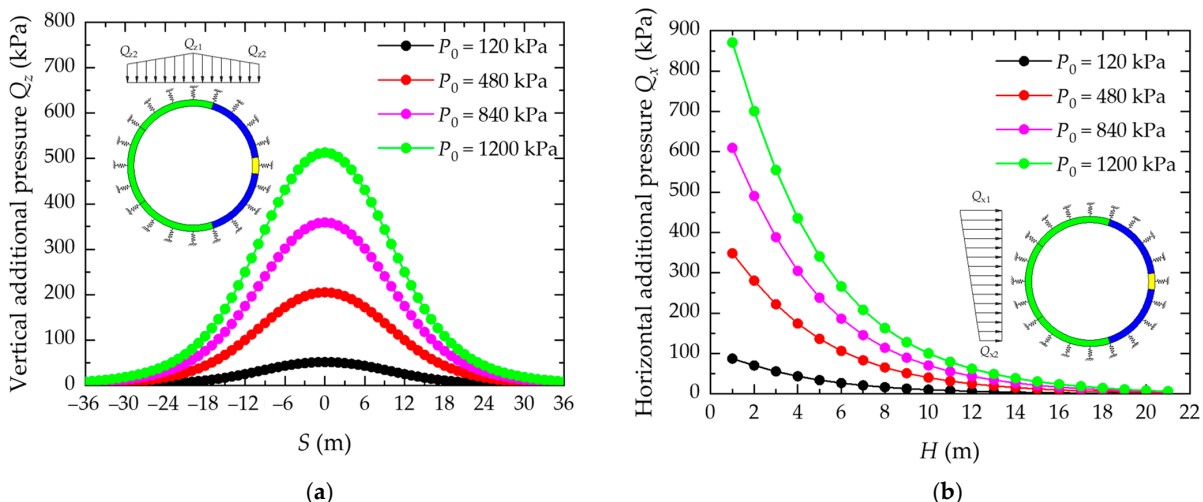

**Figure 9.** Variation curves of additional pressures under ground surcharge. (**a**) Vertical additional pressure ($H$ = 15 m); (**b**) horizontal additional pressure ($S$ = 3 m).

### 3.2. Loading Structure Method

Figure 10 illustrates the schematic of the load distribution on the tunnel under the initial earth pressure conditions according to the load structure method. This method considers several key loads: the vertical earth pressure ($P_{v1}$), the horizontal earth pressure ($P_x$), the ground reaction force ($P_{y1}$), and the tunnel structure's self-weight ($G$). The horizontal earth pressure is derived from the vertical earth pressure, multiplied by the lateral pressure coefficient ($K_0$), and it increases linearly with depth. To simulate the interaction between the ground and the tunnel segment, the study employs a circumferential spring model around the segment [33]. This model comprises three springs at each segment node: one normal and two tangential springs. The normal springs are primarily utilized to simulate the radial resistance of the strata against the tunnel during deformation processes, while the tangential springs are employed to represent the tangential frictional forces exerted by the strata on the tunnel. Notably, the normal springs are designed to compress without tension, meaning that they do not provide restraint when the contact pressure between the soil and the tunnel is zero. The ground spring's stiffness is set by the subgrade reaction coefficient, while the tangential springs' stiffness parameters are one third of the radial spring's, based on the research by Koyama [34] and Wang [35]. The study, focusing on the loading structure method's impact on shield tunnel lining, divides the force on the lining structure into two stages. The first stage involves the initial earth pressure, and the second stage includes the additional load from ground surcharge. Based on the distributions of horizontal additional pressure and vertical additional pressure shown in 9, this study simplifies the linearly varying distributions, as depicted in Figure 10b, which shows the load pattern schematic for ground surcharge conditions.

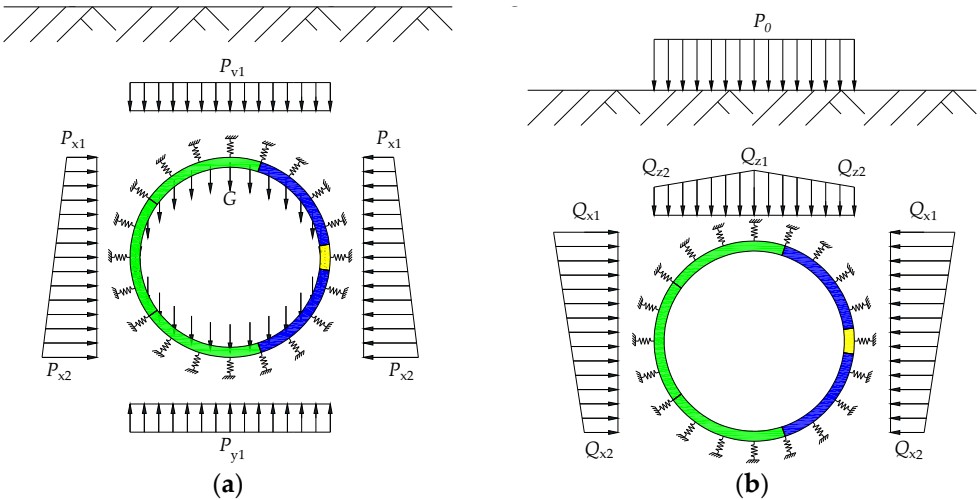

**Figure 10.** Schematic diagram of load distribution. (**a**) Normal condition; (**b**) additional pressure under ground surcharge.

### 3.3. Numerical Cases

Figure 9 illustrates the distribution of additional pressure, highlighting that the maximum occurs when the ground surcharge is directly above the tunnel. This study, therefore, concentrates on the deformation characteristics of the tunnel lining under these conditions. To analyze larger deformations, calculations were made under ten different loading conditions, varying from 120 to 1200 kPa. The shield tunnel's initial burial depth is 15 m ($H = 15$ m, $P_{v1} = 300$ kPa). The coefficients used in the analysis are the subgrade reaction ($K_r$) at 10 MPa/m and the lateral pressure ($K_0$) at 0.6 [36]. The ground surcharge is positioned directly above the tunnel ($S = 0$ m) over an 18 m-square area ($B = L = 18$ m). The specific conditions are detailed in Table 3.

**Table 3.** Calculation cases.

| Case | $P_{v1}$ (kPa) | $K_r$ (MPa/m) | $K_0$ | $S$ (m) | $B$ (m) | $L$ (m) | $P_0$ (kPa) |
|------|------|------|------|------|------|------|------|
| C1 | 300 | 10 | 0.6 | 0 | 18 | 18 | 120 |
| C2 | 300 | 10 | 0.6 | 0 | 18 | 18 | 240 |
| C3 | 300 | 10 | 0.6 | 0 | 18 | 18 | 360 |
| C4 | 300 | 10 | 0.6 | 0 | 18 | 18 | 480 |
| C5 | 300 | 10 | 0.6 | 0 | 18 | 18 | 600 |
| C6 | 300 | 10 | 0.6 | 0 | 18 | 18 | 720 |
| C7 | 300 | 10 | 0.6 | 0 | 18 | 18 | 840 |
| C8 | 300 | 10 | 0.6 | 0 | 18 | 18 | 960 |
| C9 | 300 | 10 | 0.6 | 0 | 18 | 18 | 1080 |
| C10 | 300 | 10 | 0.6 | 0 | 18 | 18 | 1200 |

## 4. Results and Analyses

The numerical model established in this study is for a shield tunnel lining structure composed of three rings assembled with staggered joints. Due to the limited number of rings, which is only three, it is impossible to eliminate the influence of longitudinal boundaries. However, the primary focus of this research is to investigate the lateral deformation characteristics of the lining structure under ground load. To mitigate the impact of longitudinal boundary constraints, the analysis mainly concentrates on the middle lining ring. The middle ring, being constrained by the adjacent rings on both sides, significantly reduces the influence of longitudinal boundary constraints. This research investigates how ground surcharge impacts the deformation of shield tunnel linings, focusing on various aspects: tunnel convergence, joint opening, bolt stress levels, concrete cracking, and stress on reinforcements.

### 4.1. Tunnel Convergence

Figure 11 illustrates the horizontal displacement pattern of the tunnel lining. The image reveals that under ground load, the lining's deformation resembles a "transverse ellipse" with its horizontal axis elongated and vertical axis compressed. As the external pressure increases, this elliptical distortion becomes more pronounced. The deformation is most noticeable at the vault, arch bottom, and haunch of the tunnel, where the joints open differently: those near the vault and arch bottom tend to open inward, while those near the haunch open outward. Figure 11c,d reveals that the lining structure becomes asymmetrical following deformation. The main reason for this is that the structural model employed in this study is a lining ring structure assembled with three-ring misaligned joints, where the position of the top piece in each ring differs. This three-ring misaligned joint assembly does not result in a symmetrical lining ring structure. Furthermore, the earth pressure acting on the structure and the additional loads generated by surface stacking are not vertically symmetrical. Therefore, the deformation of the lining ring structure should also be asymmetrical.

Figure 12 demonstrates the correlation between the horizontal convergence of the tunnel lining ($\Delta D$) and the magnitude of ground surcharge ($P_0$). The concept of horizontal convergence in tunnel engineering refers to the change in the tunnel's horizontal diameter, a commonly used metric for assessing tunnel deformation. $\Delta D$ represents the difference in tunnel diameter before and after deformation. A positive $\Delta D$ indicates an increase in tunnel diameter, whereas a negative $\Delta D$ signifies a decrease. This clarification has been added to the document for comprehensive understanding. The data indicate a direct, linear relationship: as $P_0$ increases, the horizontal convergence also increases. This pattern is primarily due to the use of constant-stiffness compression springs in the study to model the interaction between the tunnel and the surrounding soil. When the tunnel lining is subjected to ground surcharge, it deforms and the ground springs provide a restraining force. These radial springs, designed to only compress, maintain constant stiffness. Consequently, as

the tunnel deforms, the reaction force from the springs increases linearly, mirroring the linear increase in the lining's deformation.

Horizontal displacement (m)

(**a**)

Horizontal displacement (m)

(**b**)

Horizontal displacement (m)

(**c**)

Horizontal displacement (m)

(**d**)

**Figure 11.** Distribution of lining structure displacement (deformation magnified by 10 times). (**a**) Case C1 ($P_0$ = 120 kPa); (**b**) Case C4 ($P_0$ = 480 kPa); (**c**) Case C7 ($P_0$ = 840 kPa); (**d**) Case C10 ($P_0$ = 120 kPa).

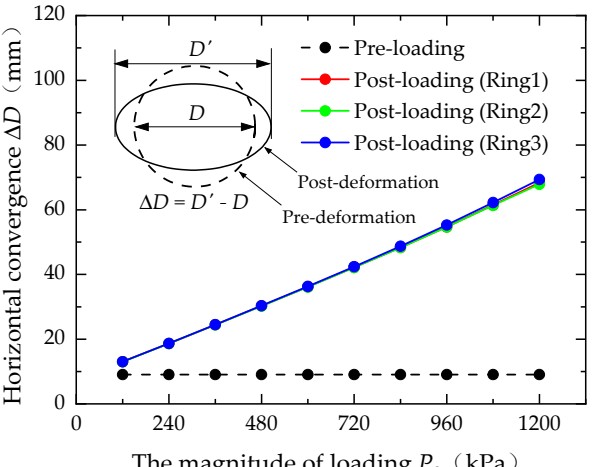

**Figure 12.** Variation in horizontal convergence with $P_0$.

### 4.2. Joint Opening and Bolt Stress

Figure 13 illustrates the changes in joint openings. It reveals that the openings at joints J1 (7.5°), J2 (72°), J3 (144°), and J5 (288°) increase significantly as $P_0$ rises. Notably, joints J1 and J2 exhibit similar expansion trends, as do J3 and J5. In contrast, the openings at J4 and J6 remain relatively stable, with all measurements under 0.2 mm. Through analyzing the deformation patterns of the lining structure, it can be observed that J1 (7.5°), J3 (144°), J4 (216°), and J6 (352.5°) demonstrate outer arc openings, whereas J2 (72°) and J5 (288°) show inner arc openings. This pattern indicates that the bolts at J1, J3, J4, and J6 experience tension on the outer side and compression on the inner side, while those at J2 and J5 undergo the opposite, with tension on the inner side and compression on the outer side.

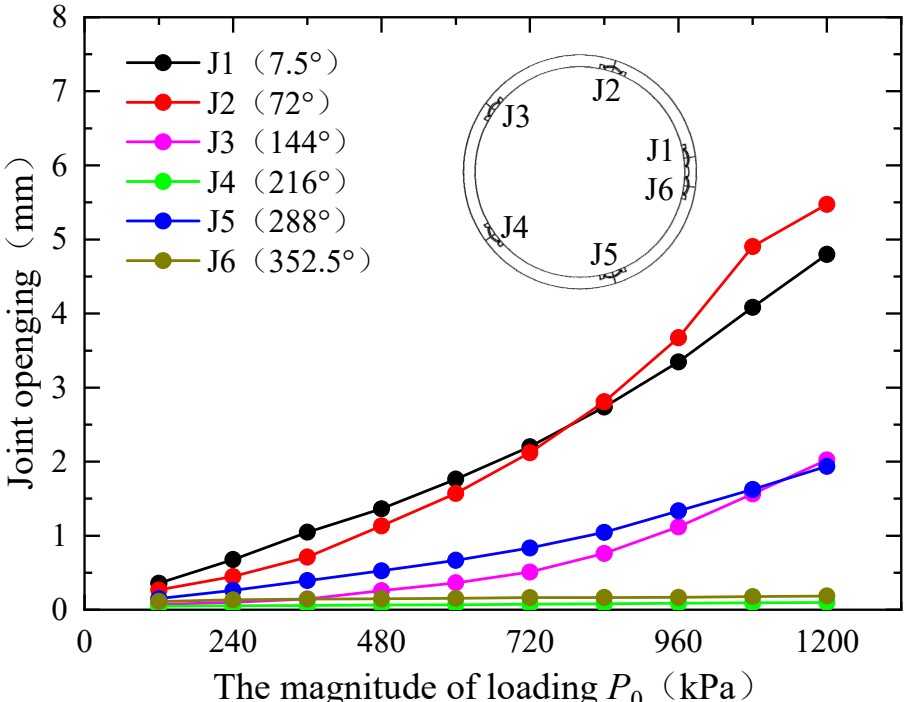

**Figure 13.** Variation in joint opening with $P_0$.

Figure 14a and b depicts the variations in bolt tensile and compressive stress, respectively, where positive values indicate tension and negative ones represent compression. Notably, J2 experiences the most significant fluctuations in both tension and compression stresses. J1 and J5 display similar stress variations, whereas J3, J4, and J6 exhibit less change. Interestingly, the maximum tensile and compressive stresses within the same bolt are nearly equal. A comparison of joint opening and bolt stress data reveals that, although J1 and J2 have similar joint openings, J2's bolt stress is higher. This suggests that larger joint openings cause bolts to exert more constraint on the inner arc surface compared to the outer arc surface. Specifically, when the joint opening on the inner arc surface reaches 4.9 mm, the bolt stress hits the yield strength of 640 MPa. Furthermore, Figure 15 illustrates the distribution of bolt stresses. Prior to ground surcharge, the maximum tensile stress on the bolts occurs at J2, and the maximum compressive stress at J6. Post ground surcharge, both the highest tensile and compressive stresses are found at J2, with the most significant stress concentrations near the nut.

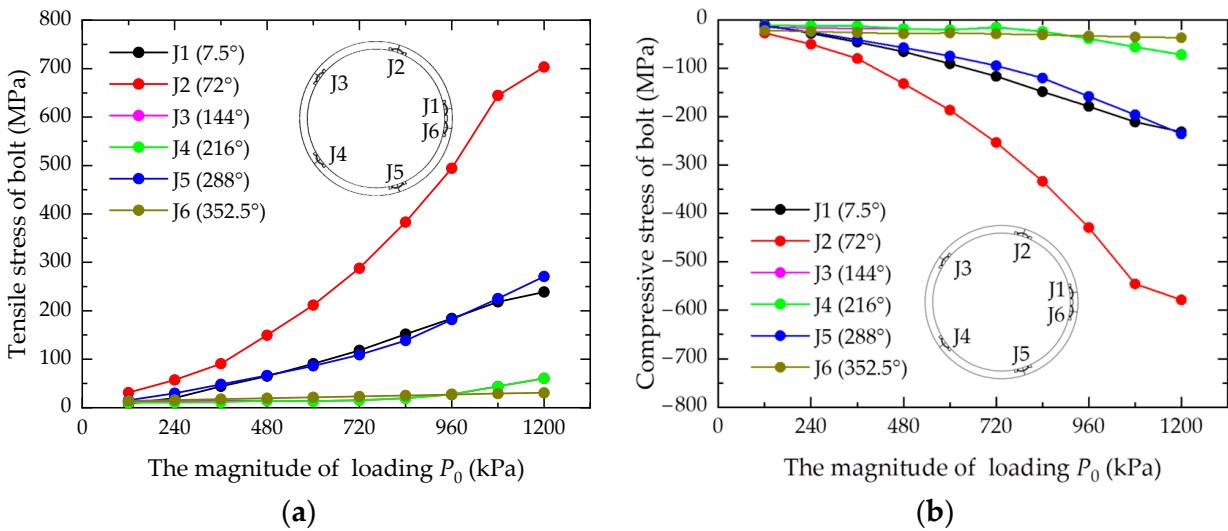

**Figure 14.** Variation in bolt stress with $P_0$. (**a**) Tensile stress; (**b**) Compressive stress.

Bolt stress (Pa)

| +1.50×10⁷ |
| +1.25×10⁷ |
| +1.00×10⁷ |
| +7.50×10⁶ |
| +5.00×10⁶ |
| +2.50×10⁶ |
| −2.50×10⁶ |
| −5.00×10⁶ |
| −7.50×10⁶ |
| −1.00×10⁷ |
| −1.25×10⁷ |
| −1.50×10⁷ |
| −1.75×10⁷ |
| −2.00×10⁷ |

○ Position of maximum tensile stress
○ Position of maximum compressive stress

(**a**)

Bolt stress (Pa)

| +6.00×10⁷ |
| +5.00×10⁷ |
| +4.00×10⁷ |
| +3.00×10⁷ |
| +2.00×10⁷ |
| +1.00×10⁷ |
| +5.00×10⁶ |
| −5.00×10⁶ |
| −1.00×10⁷ |
| −2.00×10⁷ |
| −3.00×10⁷ |
| −4.00×10⁷ |
| −5.00×10⁷ |
| −6.00×10⁷ |

○ Position of maximum tensile stress
○ Position of maximum compressive stress

(**b**)

Bolt stress (Pa)

| +1.50×10⁸ |
| +1.25×10⁸ |
| +1.00×10⁸ |
| +7.50×10⁷ |
| +5.00×10⁷ |
| +2.50×10⁷ |
| +1.50×10⁷ |
| −1.50×10⁷ |
| −2.50×10⁷ |
| −5.00×10⁷ |
| −7.50×10⁷ |
| −1.00×10⁸ |
| −1.25×10⁸ |
| −1.50×10⁸ |

○ Position of maximum tensile stress
○ Position of maximum compressive stress

(**c**)

Bolt stress (Pa)

| +3.00×10⁸ |
| +2.50×10⁸ |
| +2.00×10⁸ |
| +1.50×10⁸ |
| +1.00×10⁸ |
| +5.00×10⁷ |
| +2.50×10⁷ |
| −2.50×10⁷ |
| −5.00×10⁷ |
| −1.00×10⁸ |
| −1.50×10⁸ |
| −2.00×10⁸ |
| −2.50×10⁸ |
| −3.00×10⁸ |

○ Position of maximum tensile stress
○ Position of maximum compressive stress

(**d**)

**Figure 15.** *Cont*.

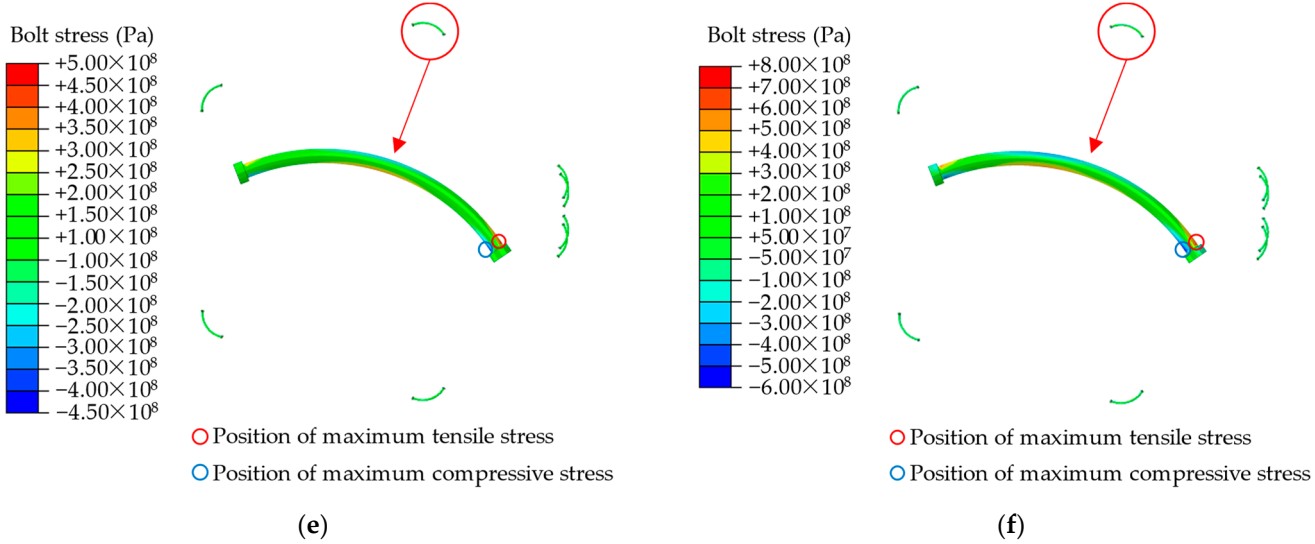

**Figure 15.** Distribution of bolt stress. (**a**) Pre-loading; (**b**) Case C2 ($P_0$ = 240 kPa); (**c**) Case C4 ($P_0$ = 480 kPa); (**d**) Case C6 ($P_0$ = 720 kPa); (**e**) Case C8 ($P_0$ = 960 kPa); (**f**) Case C10 ($P_0$ = 1200 kPa).

### 4.3. Rebar Stress

Figure 16 illustrates the correlation between reinforcement stress and $P_0$. As shown in Figure 16a, an increase in $P_0$ leads to a rise in the maximum tensile stresses in the reinforcement of segments B1, B2, B3, and L2, while segments F and L1 show minimal changes. Specifically, at $P_0$ = 720 kPa, segment B1's reinforcement reaches a maximum tensile stress of 413.66 MPa, and, at $P_0$ = 1080 kPa, segment B2's reinforcement hits a maximum of 458.05 MPa, with both matching the yield strength of the reinforcement. Figure 16b displays the progression of maximum compressive stresses in the reinforcements, indicating a consistent increase across all segments with rising $P_0$. However, the variations in maximum tensile and compressive stresses are more pronounced in segments B2 and L2, yet they do not reach the yield strength. This suggests that under ground surcharge conditions, the reinforcement in the lining structure predominantly experiences tensile stress.

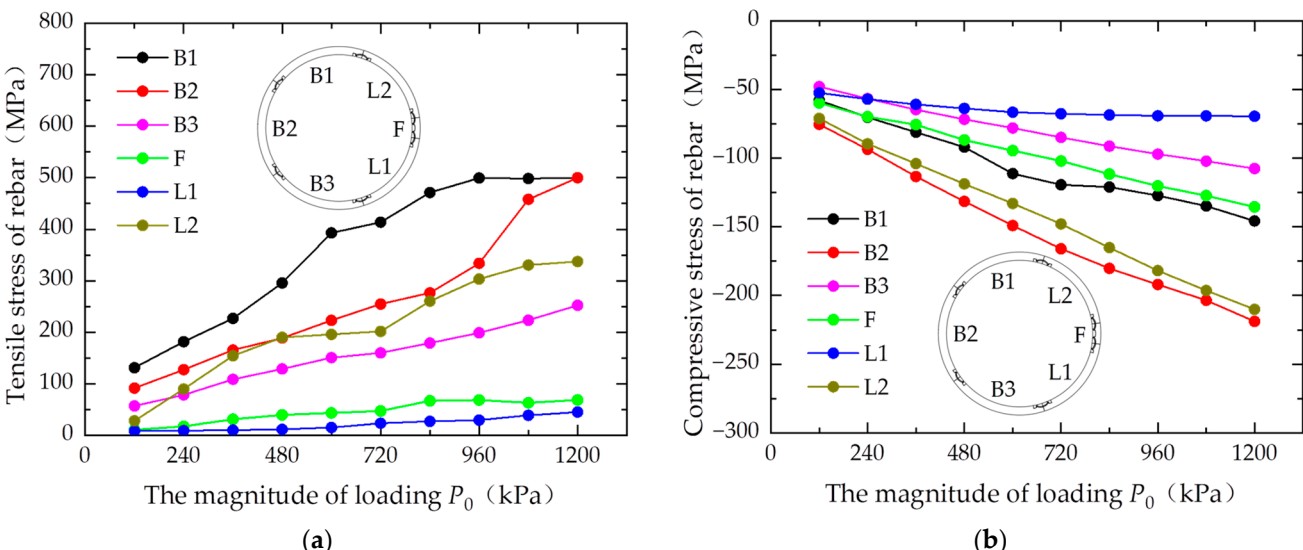

**Figure 16.** Variation in rebar stress with $P_0$. (**a**) Tensile stress; (**b**) Compressive stress.

Figure 17 presents the distribution of rebar stress, where tension is indicated by positive values and compression by negative ones. In Figure 17a, before ground surcharge, the maximum tensile and compressive stresses in segment B1's reinforcement are located

on the inner and outer sides of the vault, respectively. Similarly, in segment B2, these stresses occur on the outer and inner sides of the haunch. In segment B3, they appear on the inner and outer sides of the arch bottom. For segments L1, L2, and F, the stresses concentrate near the joints. Figure 17b shows that at $P_0$ = 240 kPa, the stress distribution in each segment mirrors the pre-surcharge condition. Figure 17c–f reveal that as $P_0$ increases, the maximum stresses in all segments are primarily around the vault, haunch, arch bottom, and joints. Notably, the maximum tensile stress in the reinforcement shifts towards the joints with increasing $P_0$. At lower $P_0$ values, the joint deformation is minimal and the segments mainly undergo bending deformation. However, at higher $P_0$ values, significant local compressive deformation at the joints results in increased tensile stress in the rebar. The maximum compressive stress, primarily a result of segment bending, also localizes near the vault, haunch, arch bottom, and joints.

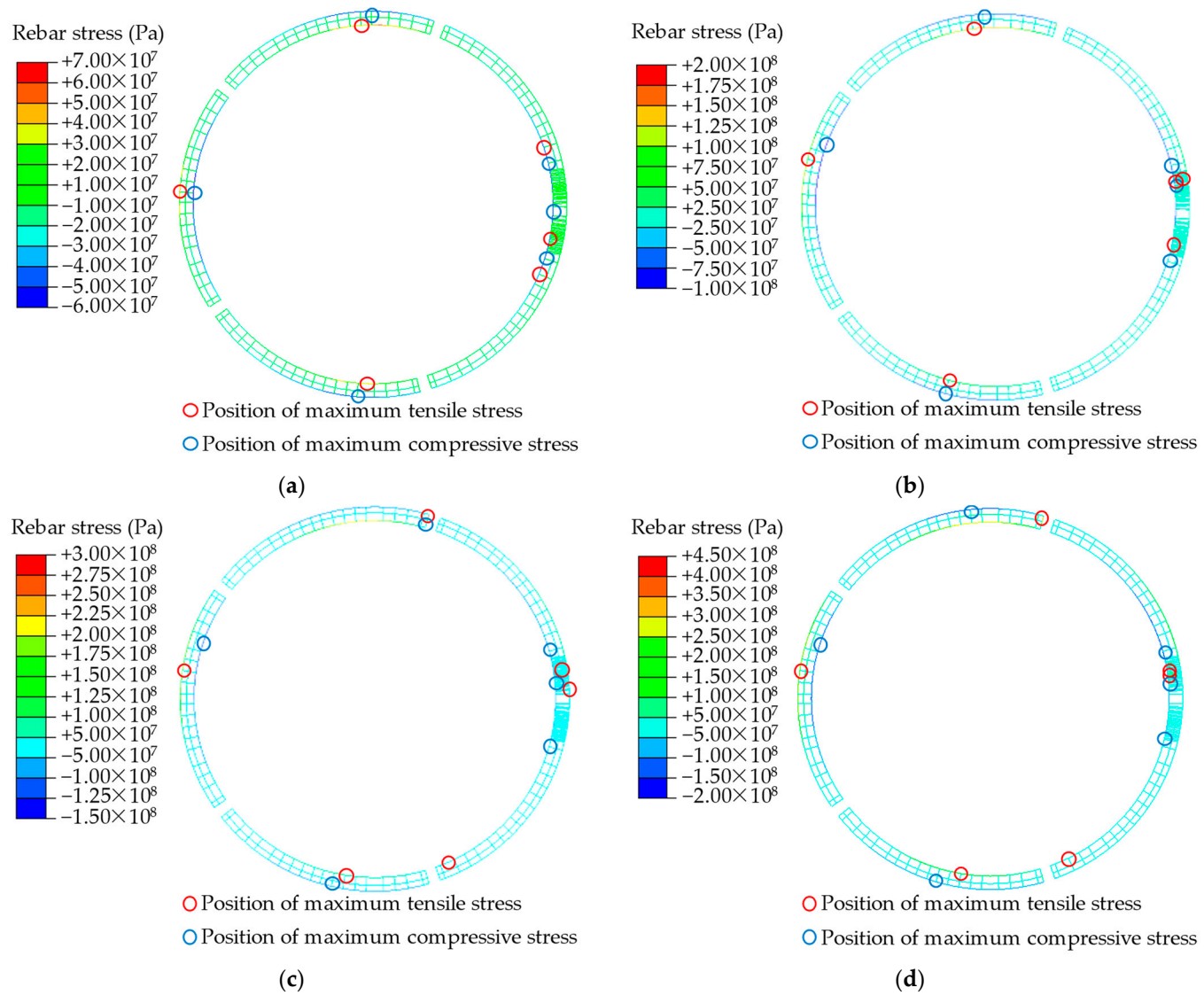

**Figure 17.** *Cont.*

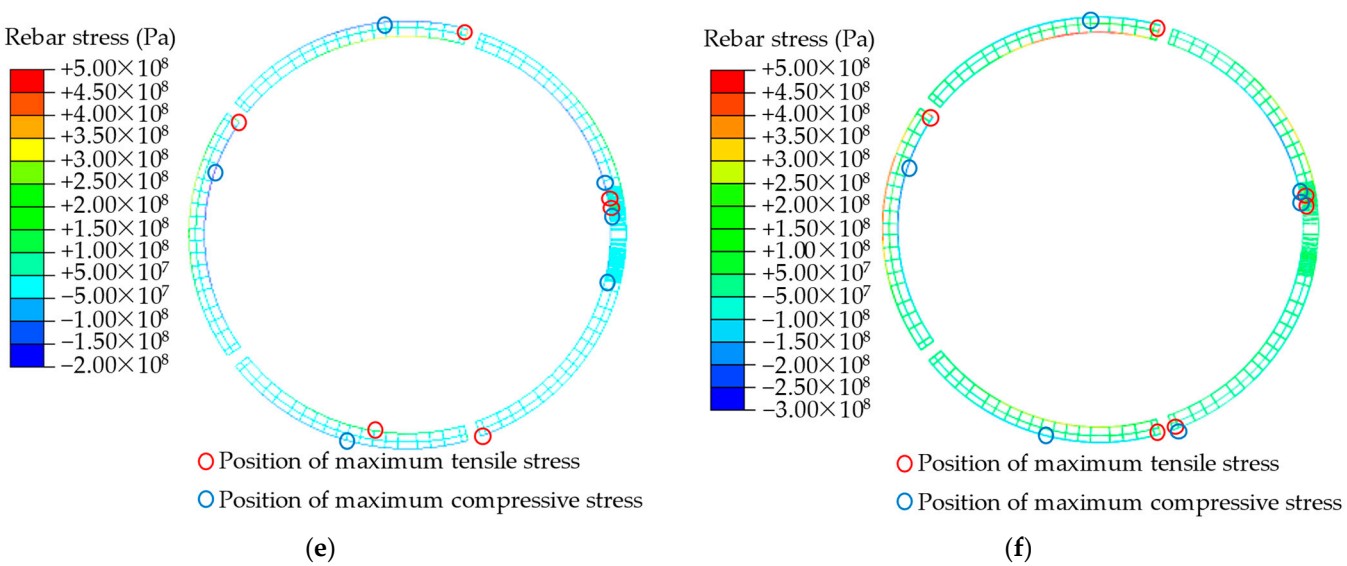

**Figure 17.** Distribution of rebar stress. (**a**) Pre-loading; (**b**) Case C2 ($P_0$ = 240 kPa); (**c**) Case C4 ($P_0$ = 480 kPa); (**d**) Case C6 ($P_0$ = 720 kPa); (**e**) Case C8 ($P_0$ = 960 kPa); (**f**) Case C10 ($P_0$ = 1200 kPa).

### 4.4. Cracking of Concrete

Figure 18 illustrates the crack distribution in the lining structure. The subfigures Figure 18a–j show the crack patterns at various $P_0$ levels, while Figure 18j provides a detailed view of the cracks at $P_0$ = 1200 kPa from multiple perspectives. The analysis of Figure 18a–j reveals that the cracks predominantly appear near the vault, haunch, and arch bottom, intensifying as $P_0$ increases both in number and extent. In Figure 18j, the most severe cracks are noted at these same locations. The crack distribution ranges are specified as follows: for the haunch (at 0° and 180°), the range is from 340° to 50° and 130° to 200°, respectively; for the vault (90°), it spans from 55° to 125°; and for the arch bottom (270°), the range extends from 225° to 310°. The central angle of these crack ranges lies between 70° and 85°.

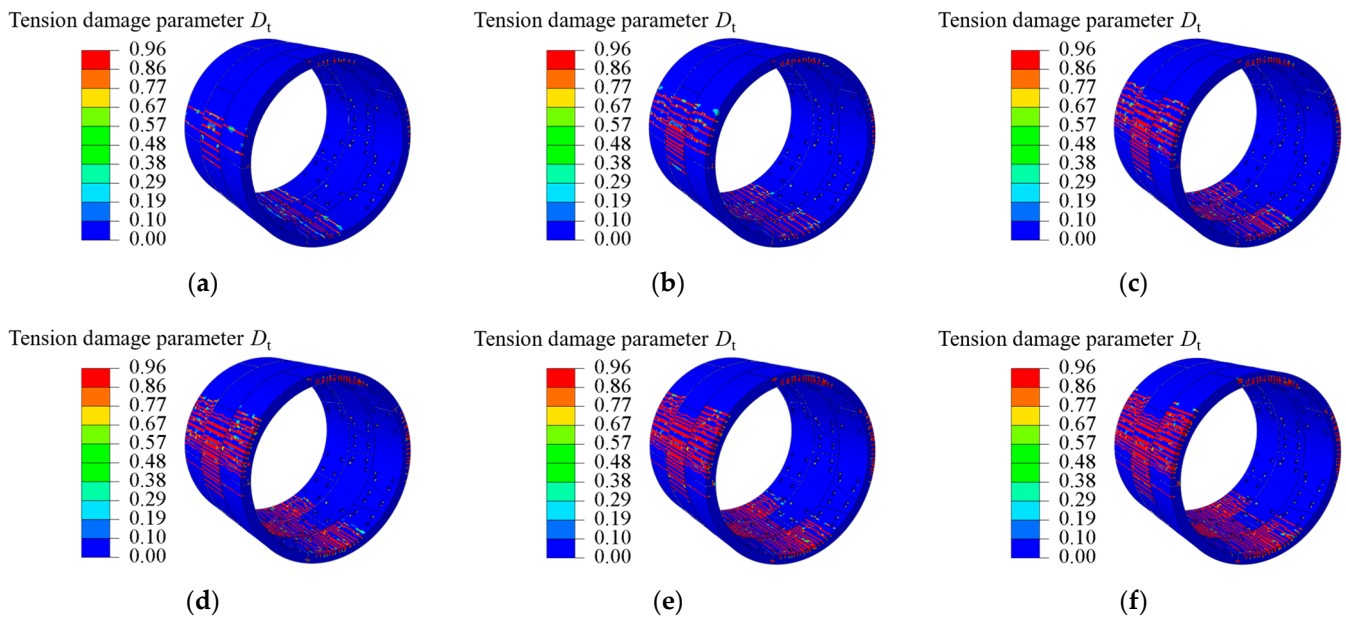

**Figure 18.** *Cont.*

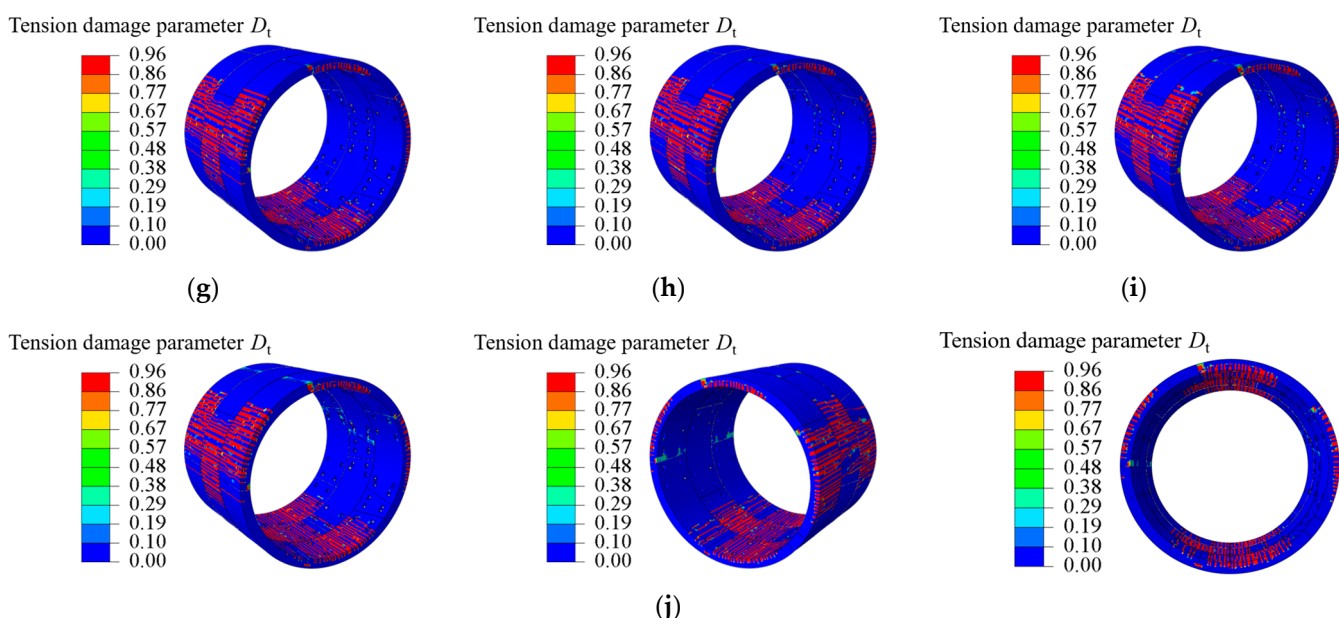

**Figure 18.** Development of concrete cracks (tension damage parameter $D_t$). (**a**) Case C1 ($P_0 = 140$ kPa); (**b**) Case C2 ($P_0 = 240$ kPa); (**c**) Case C3 ($P_0 = 360$ kPa); (**d**) Case C4 ($P_0 = 480$ kPa); (**e**) Case C5 ($P_0 = 600$ kPa); (**f**) Case C6 ($P_0 = 720$ kPa); (**g**) Case C7 ($P_0 = 840$ kPa); (**h**) Case C8 ($P_0 = 960$ kPa); (**i**) Case C9 ($P_0 = 1080$ kPa); (**j**) Case C10 ($P_0 = 1200$ kPa).

## 5. Conclusions

This study successfully developed and validated a refined numerical model for shield tunnel lining structures with staggered joints, employing finite element software and full-scale model tests to ensure model reliability. By integrating the Boussinesq formula with advanced numerical simulations, this research innovatively calculates the impact of ground surcharge, offering new insights into the deformation characteristics of shield tunnels. These findings significantly contribute to the field by providing a comprehensive analysis of tunnel convergence, internal forces, joint dynamics, and stress responses in concrete and reinforcement, thereby enhancing design and monitoring strategies in tunnel engineering. The main conclusions are as follows:

(1) This research established a numerical model for shield tunnel lining structures, meticulously comparing the convergence deformation of tunnel haunches against full-scale shield tunnel segment tests. The findings reveal an alignment in deformation trends between numerical simulations and physical tests across various loading phases, despite the presence of errors ranging from 5% in the elastic loading phase to 47% in the plastic unloading phase. These discrepancies highlight the complexity of accurately simulating structural plastic deformations. However, the consistent trend agreement underscores the model's validity and its potential as a basis for further investigation. This study not only validates the numerical model's utility in predicting the mechanical behavior of shield tunnel linings but also underscores the need for refining these models to enhance their predictive accuracy, particularly in the plastic deformation phases. Future research should aim to reduce simulation errors by exploring the impacts of different loading conditions.

(2) This study has elucidated the deformation behaviors of shield tunnel lining structures under ground surcharge, revealing a distinctive pattern of horizontal elongation and vertical compression, resulting in an "oval"-shaped deformation. The investigation into joint deformations has highlighted a critical response mechanism: segment rotation, dependent on the orientation of joint deformation. Specifically, the rotation hinge's location varies with the deformation orientation, facilitating a predictive model for the deformation mode of lining structures in practical monitoring scenarios. This insight not only enhances our understanding of shield tunnel lining behavior under external loads but also lays the

groundwork for developing more accurate predictive tools and methodologies for tunnel design and monitoring.

(3) This research delineates a direct correlation between joint deformation patterns and bolt stress within shield tunnel linings, identifying tension and compression dynamics based on the direction of joint opening. Our findings reveal that bolts experience increased stress, reaching the material's yield strength, particularly when joints open toward the inner arc. Upon the inner arc joints' opening reaching 4.9 mm, the stress exerted on the bolts escalates to the yield strength of 640 MPa, with a peak stress concentration observed near the nut.

(4) Under the influence of ground surcharge, significant deformation is observed in the lining structure, with the maximum tensile stress in the segmental reinforcement being primarily located at the joints. This occurrence is attributed mainly to the fact that, under heavy loads, tensile forces are predominantly borne by segmental reinforcement, and compression at the joints leads to significant deformation. This results in the local tensile yielding of the reinforcement at the joints. This insight is critical for tunnel design, suggesting the necessity for localized reinforcement near joints.

(5) Crack formation in the lining structure is predominantly observed near the inner arc of the vault and arch bottom, as well as the outer arc of the haunch, under the influence of ground surcharge. These cracks typically span specific angular ranges, manifesting within a central angle of 70° to 85°. In monitoring activities, it is crucial to prioritize observation in these specific areas, as they are under significant tensile stress, rendering the concrete more susceptible to cracking. Upon the occurrence of cracking, the reinforcement steel assumes the primary load-bearing responsibility. To address this issue, design strategies incorporating localized strengthening are recommended.

**Author Contributions:** Conceptualization, L.W., C.Y. and W.C.; methodology, L.W., W.C. and D.S.; data curation, C.Y. and W.C.; data analysis, L.L. and D.S.; writing—original draft preparation, W.C.; writing—review and editing, D.S.; supervision, L.W.; project administration, C.Y.; funding acquisition, L.L. and D.S. All authors have read and agreed to the published version of the manuscript.

**Funding:** This research work was funded by the Science and Technology Innovation Plan 2022 of Ministry of Housing and Urban–Rural Development of Guangdong Province (2022-K4-094918), Shenzhen Science and Technology Program (20220808143139001), and Shenzhen University 2035 Program for Excellent Research (2022B007).

**Institutional Review Board Statement:** Not applicable.

**Informed Consent Statement:** Not applicable.

**Data Availability Statement:** All data included in this study are available upon request via contact with the corresponding author. The data are not publicly available due to privacy reasons.

**Conflicts of Interest:** Author Lixin Wei, Chunshan Yang and Liying Liu was employed by the company Guangzhou Municipal Engineering Design & Research Institute Co., Ltd. (Guangzhou, China). The remaining authors declare that the research was conducted in the absence of any commercial or financial relationships that could be construed as a potential conflict of interest.

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
