# Peer review of "Numerical Analysis of Ground Surcharge Effects on Deformation Characteristics in Shield Tunnel Linings"

_applsci, doi:10.3390/app14062328_

Round 1
Reviewer 1 Report
Comments and Suggestions for Authors
The Authors have dealt with a very interesting topic, but the article requires numerous changes before it can be published.
Figure 1
Why did the Authors study only three staggered rings, instead of ten (to have a complete rotation)? Is there a reason related to calculation time? The Authors should justify their choice in the text.
Figure 2
What are the additional hand holes, which are seen in this figure but do not appear in Figure 1, for? Since they are not used to position the bolts, are they used for moving the pieces? The Authors should explain in the text.
Section 2.4
The Section should also specify what the interactions are like between the first and third rings with the rings preceding and following the 3-ring module. In other word, the Authors should specify the boundary conditions on the initial and final cross-section.
Lines 149-150
“This study’s finite element model effectively captures the friction at the interface between segments.”
In order for the numerical result to be considered representative of real behavior, many more rings would be needed (not just 3), to avoid edge effects. For a model like this that aims to capture interface friction, therefore, the question of the number of rings modeled is particularly important. The Authors should therefore highlight the limits on the reliability of the results deriving from having modeled a limited number of rings. The reviewer does not intend to question whether the model is able to capture the friction effect but only wants to point out that, from a quantitative point of view, the results may not be in line with the real values, due to edge effects.
Lines 170-171
“Wang et al. [35] conducted full-scale model tests to validate the accuracy of the numerical model used in this study.”
There were only three rings in the full-scale model? Please clarify.
Lines 172-173
“This included 6 P1 loads, 10 P2 loads, and 8 P3 loads.”
Include references to Figure 5 and Table 2, to clarify in what the loads P1, P2, and P3 differ.
Lines 185-187
“However, for conditions L3 to L4, there’s a noticeable discrepancy, primarily due to the simulation only incorporating the main loading conditions from the test”
Could edge effects play some role in driving this discrepancy?
Lines 207-208
“The normal spring activates only under nonzero contact pressure between soil and tunnel.”
What about the sign of the normal stress? Do the normal springs activate for both positive (tensile) and negative (compressive) normal stresses? You probably specified this in line 283(“These radial springs, designed to only compress”), but you should anticipate it here.
Lines 220-221
“This study calculates the additional ground surcharge load using the Boussinesq formula.”
Be more precise. Provide here the additional ground surcharge load formula.
Lines 225-226
“Figure 8 presents a scenario where a rectangular ground surcharge is positioned above the operational underground tunnel”
Above? The ground surcharge in Figure 8 is displaced laterally relative to the operational underground tunnel.
Line 235
“μ is the Poisson’ ratio of the soil”
You have already used this symbol for the viscosity parameter of concrete (Table 1). Change symbol. In Table 1, you used ν for the Poisson’s ratio. You should be consistent. A table of the symbols used at the beginning of the article might be helpful.
Lines 255-263
“Figure 9 illustrates the distribution of additional pressure, highlighting that the maximum occurs when the ground surcharge is directly above the tunnel. This study, therefore, concentrates on the deformation characteristics of the tunnel lining under these conditions. To analyze larger deformations, calculations were made under ten different loading conditions, varying from 120 to 1200 kPa. The shield tunnel's initial burial depth is 15 meters (H = 15 m, Pv1 = 300 kPa). The coefficients used in the analysis are: the subgrade reaction (Kr) at 10 MPa/m and the lateral pressure (K0) at 0.6 [42]. The ground surcharge is positioned directly above the tunnel (S = 0 m) over an 18-meter square area (B = L = 18 m). The specific conditions are detailed in Table 3.”
Repetition of lines 245-254. Choose the version to keep.
Lines 272-273
“As external pressure increases, this elliptical distortion becomes more pronounced.”
The deformed configuration in Figure 10d is not symmetric, since the 3-element model is not symmetric. By increasing the number of rings up to 10, or multiples of 10, the deformed configuration should become symmetrical. The analysis of the results should take this into account before drawing conclusions.
Lines 277-278
“Figure 11 demonstrates the correlation between the horizontal convergence of the tunnel lining (ΔD) and the magnitude of ground surcharge (P0).”
Is this actually a convergence, or a divergence?
Since the deformed rings are not perfect ellipses, how did you calculate ΔD?
Figure 11
The legend reads “Past-loading”. Is this a typo? “Post-loading” would seem more appropriate.
Line 292
“Figure 12 illustrates the changes in joint openings.”
The analysis of the results should take the limited number of rings into account before drawing conclusions.
Line 303
“Figure 13 depicts the variations in bolt stress”
The analysis of the results should take the limited number of rings into account before drawing conclusions.
Line 320
“Figure 15 illustrates the correlation between reinforcement stress and P0.”
The analysis of the results should take the limited number of rings into account before drawing conclusions.
Line 331
“Figure 16 presents the distribution of rebar stress”
The analysis of the results should take the limited number of rings into account before drawing conclusions.
Lines 350-355
“Figure 15 illustrates the crack distribution in the lining structure. The subfigures 15(a) to (j) show the crack patterns at various P0 levels, while Figure 15(j) provides a detailed view of the cracks at P0 = 1200 kPa from multiple perspectives. The analysis of Figures 15(a) to (j) reveals that the cracks predominantly appear near the vault, haunch, and arch bottom, intensifying as P0 increases both in number and extent. In Figure 15(j), the most severe cracks are noted at these same locations.”
Are you talking about Figure 17 and 17(a) to (j)?
The analysis of the results should take the limited number of rings into account before drawing conclusions.
Comments on the Quality of English Language
Minor editing of English language required.
Reviewer 2 Report
Comments and Suggestions for Authors
-
Dear Authors
I wanted to extend my heartfelt congratulations to you on your paper. Your dedication to exploring the intricate dynamics of tunnel engineering is truly commendable, and your research sheds invaluable light on a field with immense potential for the future.
Tunnel engineering plays a pivotal role in shaping our urban landscapes and transportation networks, and your commitment to delving into its complexities is both inspiring and forward-thinking. By delving into the deformation characteristics of shield tunnel linings under ground surcharge, you are contributing significantly to our understanding of this critical infrastructure sector.
Your work not only demonstrates your expertise and dedication but also underscores the importance of continued research and innovation in tunnel engineering. As we look ahead to the future, it is clear that advancements in this field will be instrumental in addressing the challenges of urbanization and improving transportation efficiency worldwide.
After carefully reviewing your paper, I believe there are a few areas where some enhancements could further strengthen the clarity and impact of your findings. Here are some suggestions:
- Line 38: The term "burgeoning development" is vague and could be clarified. Specify what aspects of urban development are burgeoning to provide more context.
-
Lines 41-42: The statement about "significant challenges to existing tunnels" lacks specificity. It would be helpful to outline these challenges explicitly to better contextualize the research problem.
-
Lines 45-47: The citation format seems incomplete. Ensure that all references are properly formatted according to the journal's guidelines.
-
Lines 53-55: The phrase "enhancing the precision and applicability of simulation methodologies" could be more specific. What aspects of precision and applicability are being enhanced, and how?
-
Lines 56-58: The list of recent studies could be condensed for brevity. Consider summarizing the key findings of these studies instead of listing them individually.
-
Line 92: The statement about "addressing the limitations of current methodologies" is vague. Specify which limitations are being addressed and how they are being overcome.
-
Lines 98-104: The description of the shield tunnel linings is detailed, but it could be more concise. Consider condensing this information to focus on the essential characteristics of the lining rings.
-
Lines 114-116: The explanation of the finite element model's elements could be clearer. Provide more context or examples to help readers understand the significance of these elements in the simulation.
-
Lines 121-123: The reference to the Concrete Damage Plasticity (CDP) constitutive model lacks clarity. Explain briefly how this model works and why it is suitable for the study.
-
Lines 129-132: The statement about the concrete's compressive and tensile strengths could be supported with more detailed information or references to relevant standards or tests.
-
Lines 146-147: The description of the reinforcement and bolts' material properties is clear, but it would be helpful to explain how these properties were determined or validated.
-
Lines 152-163: The explanation of the interactions between segments and connections could be more concise. Consider simplifying the technical details for readers who may not be familiar with structural engineering terminology.
-
Line 169-170: The reference to Wang et al. [35] for conducting full-scale model tests lacks clarity. Please provide more information about the specific study, including the title, authors, and publication details.
-
Lines 171-174: The description of the testing procedure is detailed but could be clarified further. Please specify the dimensions and materials used in the full-scale model tests to ensure reproducibility and comparability.
-
Lines 176-177: The explanation of how the numerical model replicates the full-scale test conditions needs elaboration. Please provide details on how the longitudinal load distribution was determined and how it corresponds to the actual loading conditions in the test.
-
Lines 180-181: The figures mentioned in the text (Figure 6 and Figure 7) are referenced before they are introduced. Please ensure that figures are introduced before they are cited in the text.
-
Lines 186-190: The discussion of discrepancies between simulation and test results lacks specificity. Please provide more detailed insights into the factors contributing to these discrepancies, such as material properties, boundary conditions, or modeling assumptions.
-
Lines 196-197: The figure caption for Figure 6 should specify what the comparison is based on (e.g., deformation, stress distribution) to provide clarity to readers.
-
Lines 199-200: Figure 7(a) is referenced without being introduced. Please ensure that figures are introduced before they are cited in the text to maintain coherence.
-
Lines 214-215: The description of the load pattern schematic in Figure 7(b) could be clearer. Please provide more details on how the load pattern changes under ground surcharge conditions.
-
Lines 218-220: The Boussinesq formula is mentioned, but it would be helpful to provide a brief explanation or reference for readers unfamiliar with this formula.
-
Lines 226-233: The description of the scenario for ground surcharge positioning lacks clarity. Please provide specific dimensions and parameters to help readers understand the setup more comprehensively.
-
Lines 365-370: The conclusions lack specificity and detail. It is important to provide a succinct summary of the key findings and their implications for the research field. Additionally, the conclusions should be directly related to the objectives stated in the introduction.
-
Lines 372-386: While the main conclusions are presented, they lack depth and critical analysis. The conclusions should go beyond summarizing the results and provide insights into the significance of the findings, their limitations, and potential future research directions.
-
Lines 376-377: The statement regarding the model's accuracy should be supported with quantitative evidence, such as error analysis or comparison metrics, to strengthen its validity.
-
Lines 378-381: The description of the deformation characteristics could be more detailed and supported with numerical data or visual representations from the analysis.
-
Lines 382-384: The discussion of bolt stress and reinforcement behavior under ground load lacks context. Please provide an explanation of how these factors contribute to the overall performance and safety of the tunnel lining.
-
Lines 385-386: The description of crack formation should be accompanied by insights into the implications of these cracks on the structural integrity of the tunnel lining and potential mitigation strategies.
To improve the conclusions, I would suggest:
- Providing a clear and concise summary of the main findings, directly addressing the objectives outlined in the introduction.
- Offering critical insights into the significance of the findings and their implications for the field of tunnel engineering.
- Discussing the limitations of the study and areas for future research to provide a balanced perspective on the research outcomes.
- Ensuring that each conclusion is supported by relevant data or analysis from the study.
- Clarifying the practical implications of the findings for engineering practice and infrastructure design.
Round 2
Reviewer 1 Report
Comments and Suggestions for Authors
In some cases, Authors responded to the reviewer privately and did not include the response in the text of the manuscript. The reviewer believes that including all responses in the manuscript is helpful to future readers who may have the same concerns as the reviewer.
Response 1
“The numerical model in this study was established based on a selection derived from full-scale model tests, consisting of two semi-cut lining rings (each with a width of 750 mm) and one complete lining ring (with a width of 1500 mm), with the central ring being the complete one. The choice of a three-ring structural model was motivated by the desire to validate the accuracy of the numerical model through full-scale experiments and was also influenced by computational time considerations.”
One thing is not clear to the reviewer in this explanation: if two semi-rings and one complete ring were used in the experimental tests, why is the numerical model composed of three complete rings (as it seems in Figure 2)?
In any case, the Authors should include this entire explanation in the text of the manuscript. They added a shorter explanation, but this will definitely be more interesting for future readers.
“Furthermore, this study focused on discussing changes in the lateral deformation of the lining structure, without addressing changes in the longitudinal deformation of the tunnel, hence the selection of a three-ring structural model.”
The reviewer’s observation does not concern the longitudinal deformation of the tunnel but precisely the lateral deformation of the lining structure. In fact, having only three rings does not make the configuration symmetrical, while the repetitive module with ten rings finds a sort of symmetry. This means that the three-ring structural model is not representative of the actual lateral deformations in a real portion of the lining structure. However, it is representative of the lateral deformations in the experimental tests and this is reasonable in the context of a numerical model validation, but needs to be adequately highlighted in the text. Please specify this in the text.
Response 2
“The segments include manholes for circumferential connecting bolts and longitudinal connecting bolts. While Figure 1, being a plan view, only displays the manholes for circumferential connecting bolts, Figure 2 is a three-dimensional schematic that shows both circumferential and longitudinal connecting bolt manholes. The purpose of these manholes is to facilitate the positioning of the bolts.”
The Authors should include this explanation in the text of the manuscript. It will certainly be interesting for future readers.
Response 3
“The interactions between longitudinal lining rings and between segments in the circumferential direction are identical. These interactions encompass contact between segments, between washers and segments, between nuts and washers, between bolts and bolt sleeves, as well as between bolt sleeves and segments. From the perspective of sectional schematics, they are the same.”
The Authors should include this explanation in the text of the manuscript. It will certainly be interesting for future readers.
Response 4
“the limitation on the number of rings does have a slight effect on the computational results; however, we believe this impact is substantially mitigated.”
Once again, the reviewer’s observation concerns the lateral deformation of the lining structure. The reviewer disagrees that limiting the number of rings from 10 to 3 has a slight effect on the lateral deformation. The degree of asymmetry of the three-ring structural model is too high to have only a slight effect on the numerical solution. However, it is comparable to the degree of asymmetry of the experimental tests. Therefore, this study’s finite element model effectively captures the friction at the interface between the segments of the experimental tests used for comparison with the numerical model. Please specify this in the text.
Response 11
We have made corrections in the text, standardizing the symbol for Poisson’s ratio as ν, and the symbol for the viscosity parameter of concrete as μ.
There are remaining μ symbols for Poisson’s ratio in lines 279 and 296, Equations (1) and (4).
Response 13
“The structural model employed in this study is a lining ring structure assembled with three-ring misaligned joints, where the position of the top piece in each ring differs. This three-ring misaligned joint assembly does not result in a symmetrical lining ring structure. Furthermore, the earth pressure acting on the structure and the additional loads generated by surface stacking are not symmetrical vertically. Therefore, the deformation of the lining ring structure should also be asymmetrical.”
The Authors should include this explanation in the text of the manuscript. It will certainly be interesting for future readers.
Comments on the Quality of English Language
Minor editing of English language required.
